# Calculation of the Isobaric Heat Capacities of the Liquid and Solid Phase of Organic Compounds at 298.15K by Means of the Group-Additivity Method

**DOI:** 10.3390/molecules25051147

**Published:** 2020-03-04

**Authors:** Rudolf Naef

**Affiliations:** Department of Chemistry, University of Basel, 4003 Basel, Switzerland; rudolf.naef@unibas.ch; Tel.: 41-619-119-273

**Keywords:** heat capacity, group-additivity method, ionic liquids

## Abstract

The calculation of the isobaric heat capacities of the liquid and solid phase of molecules at 298.15 K is presented, applying a universal computer algorithm based on the atom-groups additivity method, using refined atom groups. The atom groups are defined as the molecules’ constituting atoms and their immediate neighbourhood. In addition, the hydroxy group of alcohols are further subdivided to take account of the different intermolecular interactions of primary, secondary, and tertiary alcohols. The evaluation of the groups’ contributions has been carried out by solving a matrix of simultaneous linear equations by means of the iterative Gauss–Seidel balancing calculus using experimental data from literature. Plausibility has been tested immediately after each fitting calculation using a 10-fold cross-validation procedure. For the heat capacity of liquids, the respective goodness of fit of the direct (*r*^2^) and the cross-validation calculations (*q*^2^) of 0.998 and 0.9975, and the respective standard deviations of 8.24 and 9.19 J/mol/K, together with a mean absolute percentage deviation (MAPD) of 2.66%, based on the experimental data of 1111 compounds, proves the excellent predictive applicability of the present method. The statistical values for the heat capacity of solids are only slightly inferior: for *r*^2^ and *q*^2^, the respective values are 0.9915 and 0.9874, the respective standard deviations are 12.21 and 14.23 J/mol/K, and the MAPD is 4.74%, based on 734 solids. The predicted heat capacities for a series of liquid and solid compounds have been directly compared to those received by a complementary method based on the "true" molecular volume and their deviations have been elucidated.

## 1. Introduction

Most experimental measurements of thermodynamic properties, such as vaporization, sublimation, solvation, or fusion enthalpies, are usually carried out at temperatures that differ from the standard temperature, which has generally been accepted as being 298.15 K. These temperature differences lead to experimental values for the temperature-dependent properties that prevent a direct comparison of the results between various compounds or between scientific teams examining the same molecule, a deficiency which, however, can be corrected, provided that the heat capacity of the molecules under examination is known. Instead of measuring this property for a specific molecule, the large amount of experimental heat-capacity data for all kinds of compounds, such as inorganic and organic salts, liquid crystals, or ionic liquids, enabled its prediction by means of a large number of mathematical methods, a comprehensive overview of which has been given in a recent publication by the present author [1]. The majority of these prediction methods are based on the group-additivity (GA) approach, whereby the group notations vary from complete polyatomic ions as, e.g., applied by Gardas and Coutinho [2] to single atoms and their immediate neighbour atoms and ligands, as described by Benson and Buss [3]. Generally, the GA methods’ range of applicability for the prediction of any kind of descriptors varies over a large scope of molecular structures, depending on the complexity and number of the group notations as well as the number of experimental data upon which the group parameters are based. Similarly, the reliability of the predictions is highly dependent on the range of application. Zàbransky and Ruzicka [4], e.g., defined 130 functional groups including cis, trans, as well as ortho and meta corrections in the parametrization of their second-order polynomial GA model for the prediction of the liquid heat capacity and its temperature dependence, based on more than 1800 experimental data points. For the majority of compounds they reported an average deviation of below 2%. For alkanols, acids and aldehydes, however, the error was larger than 3% and rose with increasing temperature. A further limit to the use of their model was the observation that the prediction accuracy deteriorated further if the compounds contained functional groups from different families, such as *N*,*N*’-diethanolamine or 1-chloro-2-propanol. Another example of a GA method, provided by Chickos et al. [5], used 47 functional groups for the prediction calculation of the heat capacity of 810 liquids and 446 solids, reporting standard errors of 19.5 J/molK for the liquids and 26.9 J/mol/K for the solids. The authors compared these errors with the experimental uncertainties of 8.12 and 23.4 J/mol/K, respectively, which they estimated from the experimental data variations for each of 219 liquids and 102 solids published by independent sources.

A common deficiency of the GA and all the other approaches cited in [1] is that none of them enables the prediction of any specific descriptor for each and any molecular structure in the chemical realm. In the case of the heat capacities of the solid and liquid phase of molecules, however, this deficiency has been overcome in that their prediction values are determined via the "true" molecular volume (V_m_) outlined in detail in [1]. Nevertheless, this approach has encountered several other shortcomings which could not all be addressed specifically, as it is based on one single number, the molecular volume. The three most important deficiencies are 1) the general influence of the hydroxy group of alcohols and carboxylic acids, 2) the specific effects of primary, secondary, and tertiary alcohols and 3) the impact of saturated cyclic rings vs. open-chained systems on the heat capacities. Accordingly, a first attempt of a linear correlation calculation in [1], which included the molecular volume and the experimental liquid heat capacity C_p_(liq) of the complete set of compounds, for which both data were available, and which neglected the mentioned shortcomings, yielded a rather large standard deviation of 27.84 J/mol/K and a mean absolute percentage deviation (MAPD) of 8.23%. The neglect of the hydroxy-group effect on C_p_(liq) was immediately manifest in that the predicted values for all those compounds carrying at least one OH group were systematically well below the experimental ones by up to ca. 130 J/mol/K. This general deviation, obviously caused by the formation of intermolecular hydrogen bridges between the OH groups, has been considered in subsequent calculations in that the complete set of compounds was separated by means of a few simple steps in the computer algorithm into three subsets, i.e., one encompassing all molecules lacking any OH group, a second one consisting of those carrying one OH group and a third one comprising those having more than one OH group. For each of these subsets, a separate linear correlation calculation had to be carried out yielding three sets of linear parameters for the prediction of the liquid and three for that of the solid heat capacities. In this way, the first one of the mentioned shortcomings has been eliminated, which correspondingly resulted in significantly better compliance of the predictions with the experimental data. The corresponding statistical results will be discussed and used for comparison in a later section. The remaining two deficiencies concerning the various alcohol classes as well as that of cyclic vs. open-chained structures in a saturated system, which exhibit a minor but still systematically negative influence on the prediction quality, has been plausibly explained, but a reasonably straightforward treatment within the context of the V_m_ method was not feasible.

Therefore, the question arose as to whether and how well a GA approach would overcome the remaining shortcomings of the V_m_ method and enable a more accurate and reliable prediction of the heat capacities of molecules in their liquid and solid phase at the standard temperature, in awareness of the disadvantage that it would not be able to cover each and every possible compound. A particularly versatile GA method, outlined in [6], enabling in a single sweep the calculation of 14 thermodynamic [6,7], solubility- [6,7,8], optics- [6], charge- [6], environment-related [6], and physical [8,9] properties of a nearly unlimited scope and size of molecular structures should best serve this purpose, all the more so as in most cases it in principle also opened a simple means for their reliable calculation on a sheet of paper. Accordingly, the present work puts a special focus on the effects of the hydroxy groups and the cyclization of saturated molecular parts on the heat capacities and how to deal with them. The statistical results of the present GA method will be put in relation to those of the V_m_ method but also to those of the GA approach of Chickos et al. [5], as this approach can be viewed as most closely related to the present one.

## 2. Method

The present study is founded on a project-owned and regularly updated, object-oriented database of more than 32000 molecules encompassing pharmaceuticals, plant protection, dyes, ionic liquids, liquid crystals, metal-organics, lab intermediates, and many more, all of which are stored as geometry-optimized 3-dimensional structures, including—besides several further descriptors—a set of 1176 experimental heat capacities of liquids and a corresponding set of 802 heat capacities of solids.

The details of the present atom-group additivity method and the evaluation of its group contributions have been outlined in an earlier paper [6]. Accordingly, its group notations have the same meaning as that exemplified in Table 1 of [6]. However, in order to include ionic liquids for which the experimental heat capacities are known, the list of group notations has been extended by ionic atom groups representing their charged fragments, as listed in the present Table 1. These special atom groups have already successfully been utilized in the calculation of the molecules’ viscosity [8] and surface tension [9], applied in the same way as the remaining groups. For the interpretation of the ionic atom groups of Table 1, the reader is invited to read section 2 of papers [8] and [9].

In the course of the first preliminary group-contribution calculations, whereby tentatively certain “standard” atom groups have been replaced by refined ones and special groups, which will be described in the following, have been added or omitted, their statistical results quickly revealed significant improvement of the predictive quality if the groups listed in Table 2 are included in the prediction of both the liquid and solid heat capacities.

In the discussion of the shortcomings of the molecular volume-based calculations of the heat capacities outlined in the introductory section, the hydroxy group appeared to be the most accountable group for large deviations between experimental and predicted heat-capacity values, even within the restricted set of OH-containing compounds, i.e., after their separation from the remaining ones. It turned out that the definition of the OH group on saturated carbon as in ordinary alcohols by the simple atom type “O” and its neighbours “HC” was inadequate for heat-capacity calculations, in contrast to the calculations of all the other descriptors mentioned in our earlier papers [6,7,8,9]. As a consequence, an additional procedure had to be integrated in the general GA algorithm outlined in [1], which redefined the atom type “O” into “O(prim)”, “O(sec)”, or “O(tert)”, depending on the number of carbon atoms attached to the C atom neighbouring the O atom, according to the definition of primary, secondary, and tertiary alcohols, as shown in Table 2. (Consequently, the definition of their neighbourhood “HC” was no longer relevant and was thus not examined.) This redefinition procedure is only invoked if the redefined atom types appear in the group-parameters table, as a consequence of the algorithmic procedure determining that it is the content of the group-parameter tables that defines which group parameters are to be evaluated for the corresponding descriptors calculations (as explained in subsection 2.2 of [1]), and since none of the other descriptors in [6,7,8,9] requires this redefinition, this procedure is only called up for the evaluation of the group parameters of present Table 3 and Table 7 and the subsequent heat-capacity predictions. The remaining hydroxy groups attached to unsaturated carbon found in carboxylic acids and phenols are notated separately by the atom type “O” and the neighbourhood “HC(pi)”, as defined in [6].

Another point of weakness discussed in the introductory section rested in the observation that the V_m_ approach systematically scored badly in the prediction of the heat capacities of molecules with cyclic saturated moieties. This deficiency has been resolved in the present GA method in that the endocyclic single bonds in a molecule are counted and their sum multiplied by the contribution value of the special group “Endocyclic bonds” to yield the effect of the cyclic moieties in a molecule on its heat capacity. The groups “Angle60”, “Angle90”, and “Angle102” serve as corrective elements for small rings. Not surprisingly, these special groups, which take account of an effect influencing the freedom of intramolecular motion, have also successfully been applied in the prediction of the entropy of fusion [7].

The special group “(COH)n” had to be introduced in the C_p_ calculations in order to compensate for deviations found for polyols and polyacids. This special group has played its useful part already in the calculation of the surface tension [9]. The test calculations also revealed a very strong influence of intramolecular hydrogen bonds on the liquid heat capacity, which had to be taken into account by the introduction of the special group “H/H Acceptor”, a group that has also been used successfully used in the prediction of the toxicity [6], the heats of solvation, and the sublimation, vaporization, and entropy of fusion [7].

The procedure for the evaluation of the atom-group contributions, as explained in [6], is identical for the two group-parameter sets for the prediction of the heat capacities of both the liquid and solid phases and may be summarized as follows: in a first step, a list of all the compounds, for which the experimental C_p_ values are known, is extracted from the database. In the next step, each “backbone” atom (i.e., each atom bound to at least two immediate neighbours) within each molecule has an atom type and its neighbourhood assigned to by means of two character strings defining an atom group, following the rules defined in [6] (e.g., “C sp3” and “H2CO” for the C_1_ atom in ethanol) and then this group’s occurrence in the molecule is counted. The list of M molecules and their N atom groups plus their experimental values are then entered into an M × (N + 1) matrix, wherein each matrix element (i,j) receives the number of occurrences of the *j*th atomic or special group in the *i*th molecule. The normalization of this matrix into an Ax = B matrix and its balancing by means of the Gauss–Seidel calculus, e.g., according to E. Hardtwig [10], yields the atom-group contributions. This mathematical approach is based on the assumption that the prediction value of a molecule’s descriptor in question can be evaluated by simply summing up all the group contributions in the molecule. For the evaluation of the heat capacities in this study, Equation 1 has been adopted, wherein C_p_ is the heat capacity at 298.15 K, *a_i_* and *b_j_* are the group contributions, *A_i_* is the number of occurrences of the *i*th atom group, and *B_j_* is the number of occurrences of the *j*th special group.
(1)$$C_{p}=\sum_{i}a_{i}A_{i}+\sum_{j}b_{j}B_{j}$$

The reliance of this procedure is immediately examined by a subsequent 10-fold cross-validation plausibility test, carried out in a way to ensure that each compound has been entered into the calculation as a test as well as a training sample. All the group contribution values and the statistical results of both the direct equalization and the cross-validation calculation of the liquid heat capacity C_p_(liq,298) are then collected in Table 3 and for the solid heat capacity C_p_(sol,298) in Table 7. However, for the evaluation of the statistical results, only those group contributions are considered as valid for use that have been represented by at least three independent molecules in the equalization calculation. The number of molecules responsible for the respective group contribution is listed in the rightmost column of Table 3 and Table 7. Evidently, for several atom groups, this number falls short of the validity requirement. Nevertheless, as this work is part of a continuous project, these groups have deliberately remained in the parameters’ tables for future use. They might also motivate readers working in this area to contribute corresponding experimental data. In order to achieve reliable contribution values for the atom and special groups, it was necessary to filter out compounds with C_p_ values that deviated too far from the predicted results. In the present work, the limit was defined as three times the cross-validated standard deviation *q*^2^. The corresponding outliers have been excluded from the parameters’ calculations and are collected in an outliers list. The present calculations are generally restricted to molecules containing the elements H, B, C, N, O, P, S, Si, and/or halogen.

## 3. Sources of Heat-Capacity Data

The present work is essentially based on the comprehensive list of experimental heat-capacities collected in [1], used to substantiate the feasibility of the V_m_ approach. However, a recent scan of the literature brought forth a number of further publications, which either confirmed previous data or even improved the conformance with prediction, but also enabled an extension of the applicability of the present GA approach. A number of new experimental C_p_ data have been published for saturated and unsaturated hydrocarbons, especially for bicyclo[2.2.2]octane and bicyclo[2.2.2]octene [11], 1-octyne and 4-octyne [12], biphenyl [13], benzo[b]fluoranthene, benzo[k]fluoranthene, indeno[1,2,3-cd]pyrene [14], and adamantane [15]. In addition, further C_p_ values have been found for the amines hexamethylenetetramine [15], tetra-*N*-phenylbenzidine and 4,4′-bis (*N*-carbazolyl)-1,1′-biphenyl [16], 1,3,5-triazine [17], the ethers 1,3,5-trioxane [17], diethylene glycol n-pentyl ether [18], triethylene glycol monopentyl ether [19] and diphenyl ether [20], several alcohols and aldehydes [21], derivatives of glycidol [22], carboxylic acids [23,24], aliphatic esters [24,25,26,27], benzoates [28], haloalkanes [29,30], haloaromatics [31,32], thio ethers, sulfones and sulfoxides [33,34], alkanolamines, [35,36,37] and nitriles [38] For several compounds, more recent publications have been found the following: 4-ethylmorpholine [39], methionine [40], theophylline, and caffeine [41], as well as for (-)-menthone, (+)-pulegone, and (-)-isopulegol [42]. Beyond these, experimental C_p_ values of a number of new compounds have been published: namely 3-amino-4-amidoximinofurazan [43], 3-fluoro-5-(3-pyridinyloxy)- benzenamine and *N*-[3-fluoro-5-(3-pyridinyloxy)phenyl]-*N’*-3-pyridinyl urea [44], 7-Methyl-1,5,7-triazabicyclo[4.4.0]dec-5-ene [45], 2-[(4-nitro-benzoyl)-hydrazone]-propionic acid [46], eugenol and (+)-carvone [47], indapamide [48], and the explosives EDNH and DNTA [49]. For some silicon-containing compounds, new C_p_ data have been found in [50]. Finally, a few further heat capacities data of ionic liquids [51,52,53] have been included in the present dataset.

## 4. Results

### 4.1. General Remarks 

In the subsequent figures, the results of the cross-validation calculations have been superimposed in red over the training data drawn in black.The complete lists of compounds with known heat capacities used in this study are available as SDF files in the Appendix A, downloadable by external chemistry software. In addition, the Appendix A provides the results lists containing the molecules’ names and experimental, training, and cross-validation data. Beyond this, the lists of outliers of both heat-capacity calculations are also available in the Appendix A.

### 4.2. Heat Capacity of Liquids

In Table 3, the atom groups and their contribution for the prediction of the heat capacity of liquids are collected, together with the number of molecules and occurrences upon which each of them is based. 

In rows A to H, at the bottom of the table, the statistical data of this table have been gathered. As shown in row A, the group contributions have been evaluated on the basis of 1176 compounds yielding the data for 211 atom groups, of which, however, only 134 are considered as valid, i.e., that are supported by at least three compounds. Accordingly, since only valid groups have been used for the statistical evaluations, the numbers of compounds entered in the calculations of the trained and cross-validated correlation coefficients (“goodness of fit”) *r*^2^ and *q*^2^ (rows B and F) are lower with 1111 and 1060, respectively. Both the standard deviations of the complete data set (row D) as well as that of the combined cross-validation sets (row H) reveal excellently low values (in J/mol/K), not only in relation to the large range of experimental values of between 81.92 (methanol) and 1849 J/mol/K (trimethylpropane trioleate), but particularly also in comparison with the standard error of 19.5 J/mol/K reported by Chickos et al. [5] for 810 liquids. The result is a very low scatter along the correlation line, as is shown in Figure 1. Accordingly, the error distributions of both the training and the cross-validated sets fairly well follow the Gaussian distribution function, as demonstrated in the histogram (Figure 2). The MAPD for the complete set of 1111 liquid compounds was 2.66%, clearly by far better than the 8.23% for the entire set of 1303 liquids resulting from the V_m_ method [1], and still much better in comparison with the 6.51% for the OH-free subset of 1102 liquids reported in Figure 2 of [1]. 

The distinctly better conformance of the predicted with the experimental C_p_(liq) values in comparison with earlier literature references is essentially based on three primary reasons. The first one is the refinement of the molecules’ description itself by the most detailed classification of group notations, which is precluded on principal to the V_m_ method [1], but requires a large number of atom groups and consequently a large amount of experimental data for their parametrization. The second reason originated from an observation made in [1], namely that the heat capacities of primary, and less so, of secondary alcohols have notoriously been overestimated by the V_m_ approach. These systematic deviations can be seen in Table 4, where the experimental C_p_(liq,298) data and the predicted values of both the present GA and V_m_ method of the corresponding alkanols, encompassing saturated alkyl mono-, di- and polyols, are compiled for comparison. In order to overcome this deficiency, the alcohols have therefore been subdivided as described in Section 2 into the three subclasses primary, secondary, and tertiary alcohols. This additional separation indeed had a dramatically positive effect on the entire alcohols class, demonstrated by the comparison of the correlation diagrams of Figure 3. The MAPD values shown at the bottom of Table 4 confirm that the GA method on average produces distinctly lower deviations from experimental values than the V_m_ approach. 

A quick review of the contributions of the corresponding atom groups representing the primary, secondary, and tertiary alcohols (group numbers 166 to 168) in Table 3 reveals the large influence of the immediate neighbourhood of the OH group. Evidently, with its growing bulkiness, the contribution to the heat capacity of the OH group increases due to its progressively hampered accessibility to build a hydrogen bridge. This effect has been plausibly explained by Huelsekopf and Ludwig [54], who discovered, upon applying theoretical calculations based on the quantum cluster equilibrium theory (QCE) on two primary (ethanol and benzyl alcohol) and a tertiary alcohol (2,2-dimethyl-3-ethyl-3-pentanol), that primary alcohols on principle form cyclic tetramers and pentamers in the liquid phase, while tertiary alcohols under the same conditions only consist of monomers and dimers. Following this reasoning, the higher liquid heat capacity of secondary and tertiary alcohols over that of their primary counterparts having the same molecular formula is the result of their formation of smaller clusters, which inherently exhibit a higher number of rotational and translational degrees of freedom.

The third reason for the good compliance of the present C_p_ predictions with experimental values is the consideration of the cyclization effect in the present GA method. Table 5 presents a selection of some linear alkanes and their closely related cycloalkanes and compares their experimental C_p_(liq) values with predicted data calculated by means of the present GA method and the V_m_ [1] approach. Scanning the table’s fifth column immediately reveals that the V_m_ approach systematically overestimates the liquid heat capacity of the cycloalkanes, whereas those of the linear alkanes are excellently well predicted. The reason is obvious: cyclization reduces the number of rotational degrees of freedom, an effect which is categorically excluded from consideration by the V_m_ method. The present GA method, however, includes this effect in that the number of endocyclic single bonds is counted and their count is multiplied by the assigned special group contribution, in this case by the value of −3.92 J/mol/K of group 212 in Table 3. The result of this inclusion is evident in column 3 of Table 5, proving that the overestimation of the C_p_(liq) values of the cycloalkanes on average is completely lifted.

An exemplary implementation of these findings may be provided in the calculation according to Equation 1 of the C_p_(liq,298) value of a cyclic alcohol, such as cyclohexanemethanol: 

5 x [C sp3/H2C2] + [C sp3/HC3] + [C sp3/H2CO] + [O(prim)/HC] + 6 × [endocyclic bonds] = 5 × 30.06 + 21.11 + 73.79 + 14.41 + 6 × (−3.92) = 236.09 J/mol/K (experiment: 236.5 J/mol/K).

In this context, it is worth mentioning that Chickos et al. [5] took great care about the parametrization of the "cyclic tertiary sp^3^ carbons" (as they called them) and their neighbourhood, but only reserved a single atom group for all the alcohol classes including phenols.

Since, in recent years, the class of ionic liquids (IL) has received increasing interest as a group of new polar solvents, their heat capacity as an important property has come into focus. It was therefore interesting to examine how well the present GA method would cope in comparison with the V_m_ method of [1]. In Table 6, the experimental C_p_(liq,298) data of 122 ILs have been collected and compared to the prediction data calculated by the present GA method and by the V_m_ method. A comparison of the MAPD values at the bottom of the table clearly demonstrate a substantial improvement of the present GA approach over the V_m_ method.

The present calculation of the atom-group parameters for the prediction of C_p_(liq,298) revealed ca. 170 compounds with experimental values exceeding the deviation limit, as defined in Section 2, which have been removed from parameters calculations and are collected in an outliers list. A comparison of this list with that resulting from calculations by means of the V_m_ method [1] showed very high overlap, indicating that the exclusion of these compounds was indeed justified. After the removal of these outliers, a limited number of 1202 compounds with usable experimental data remained, supporting the contributions of 134 atom and special groups valid for prediction calculations, as is shown in row A of Table 3. Despite this fairly low number of atom groups, the range of applicability of the present GA method is considerably high: for nearly 62% of ChemBrain’s database of the more than 32000 compounds, the liquid heat capacity has been evaluable.

### 4.3. Heat Capacity of Solids

While the measurement of the heat capacity of liquids principally implies a consistent isotropic phase, the corresponding examination of solids very often faces the question as to what type of association the particular compound has adopted in its solid phase. Many compounds precipitate in various crystalline forms, depending on the precipitation conditions, each of them having a different heat capacity, and many of these can change from one into another crystalline structure upon measurement, perhaps even switching from one tautomeric form into another one. In some cases, the apparent solid is merely a supercooled melt. The uncertainty of the actual structure of the solids appears to be the main cause of the larger scatter of the heat capacities of solids C_p_(sol,298) as compared to that of the liquids, not only over the complete range of available compounds but also over particular compounds examined by several independent sources, as has been observed by Chickos et al. [5]. These uncertainties are expressed in the statistics data at the bottom of Table 7, which presents the list of atom and special groups and their contribution for the prediction of the heat capacity of solids. Based on the C_p_ data of 804 solids, the Gauss–Seidel calculus yielded 126 atom and special groups (row A in Table 7) valid for prediction calculations and a cross-validation standard deviation Q^2^ of 14.23 J/mol/K (row H). This standard deviation is clearly higher than that for the calculation of the liquid heat capacities, but much lower than the 26.9 J/mol/K of Chickos’ method [5] and even lower than the experimental variation of 23.4 J/mol/K for each of the 102 solids originating from independent sources [5]. The MAPD value for the complete set of solids was calculated to 4.74%, which is better than that of each of the subsets of compounds calculated by means of the V_m_ method [1]. Nevertheless, as is demonstrated in the corresponding diagram (Figure 4), the scatter around the correlation line is significantly larger compared to the one of Figure 1 for the liquid heat capacity. Analogously, the histogram (Figure 5) shows a wider "waist" than that of Figure 2.

Hydrogen bridges are known to play a crucial role in the formation of the crystalline structure of solids (think of snowflakes or water ice). Since the V_m_ approach of [1] is not able to include this effect directly, compounds containing OH groups were treated separately from the OH-free molecules. In analogy to the observation made with the liquid alcohols one would then have expected that the V_m_ approach again exhibited an unresolvable deficiency as concerning the deviations between experimental and predicted solid heat capacities of primary, secondary, and tertiary alcohols. Unfortunately, however, the enhanced extent of the scatter of the experimental C_p_(sol) values in this compound’s class concealed these suspected deviations. The present GA method on the other hand provided an indirect proof of the influence of the immediate neighbourhood of the OH group in the alcohol subclasses: a comparison of the contributions of the atom groups 157, 158, and 159 in Table 7 (−23.36, −16.25 and −3.34 J/molK, respectively), assigned to the primary, secondary, and tertiary OH groups, immediately reveals that the primary alcohols exhibit the strongest hydrogen bridge effect, leading to the correspondingly largest decrease in the heat capacity due to the additional loss of freedoms of motion, followed by the secondary and the tertiary alcohols. The reason for this differentiation is the same as explained for that of the liquid alcohols in the prior section: the increase in the bulkiness around the OH group increasingly prevents hydrogen-bridge building. The separation of the alcohol subclasses in the present GA method also improved the reliability of the predicted C_p_(sol,298) values. In Table 8, the results of the GA and the V_m_ method for 31 alkanols have been collected and compared with their experimental data. It is interesting to see that the largest deviations of the GA method coincide with large ones of the V_m_ method (i.e. for 2,2-dimethyl-1,3-propanediol and 1,15-pentadecanediol), indicating that their experimental values are probably incorrect. General experience suggests that, in cases where the V_m_ method exhibits a large deviation, it is the GA method that is more trustworthy.

While the intermolecular interactions of OH groups exhibit a large influence on the heat capacity of solids, a similar effect of saturated cyclic structures over non-cyclic ones should not be expected as their interactions merely result from the weak dispersive forces. Beyond this—and in contrast to the conditions in the liquid phase—in a solid crystal not only the translational but also the intramolecular freedoms of motion are largely restricted independent of cyclic or non-cylic molecular moieties. This seems to be confirmed by the smaller contribution of the saturated endocyclic bonds (special group 194 in Table 7) of −1.44 J/mol/K compared to that for the calculation of the liquid C_p_ of −3.92 J/mol/K. However, as has been demonstrated by the comparison of some structurally closely related examples in [1], e.g., o-, m-, and p-quinquephenyl, anthracene, phenanthrene, and various dimethylnaphthalenes, although aromatics, the chemical structure of a molecule itself has a very dominant effect on the crystalline structure, which again affects the experimental value of the solid heat capacity. In Table 9, a selection of saturated alkanes and cycloalkanes has been listed and their experimental solid heat capacities compared with the prediction values calculated by means of the present GA and the V_m_ method [1].

A quick scan of the deviations of the V_m_-calculated C_p_(sol,298) values (column 5 in Table 9) immediately shows that the V_m_ method systematically overestimates the solid heat capacity of the cycloalkanes (norbornane and the dicyclohexyldodecanes being exceptions), whereas that of the ring-free alkanes is systematically underestimated. Although the overestimation in the case of the cycloalkanes resembles the one found in the estimation of their liquid heat capacity, as demonstrated in Table 5, it does not seem far-fetched to assume that at least part of its extent lies in potentially more clearly defined crystalline structures as compared to the probably waxy consistence of the linear counterparts. The predicted C_p_(sol,298) values resulting from the present GA method, on the other hand, yield excellent conformation with the experimental data. The largest deviations are interestingly found for norbornane, one of the exceptions in the V_m_ calculations, and bicyclo[3.3.3]undecane. For norbornane, the experimental value published by Steele [55] should be higher by ca. 8% to fit the respective deviations into the general picture of both prediction methods. For bicyclo[3.3.3]undecane, both prediction methods suggest a ca. 10% higher C_p_(sol,298) value than reported by Parker et al. [56].

In conformance with the findings of the logP analysis in an earlier paper [6], the amino acids are assumed to exist in a zwitterionic form as solids (phenylglycine being an exception, as shown in Table 9 of [6], due to the lower basicity of the nitrogen atom conjugated to the phenyl ring). Accordingly, in Table 4, their carboxylate group is represented by entry 74, their alkyl- and dialkyl-ammonium functions by entries 148 and 149, respectively, and their immediate neighbours, the methyl and methylene groups, by the respective entries 4 and 11. Test calculations based on their non-ionic forms resulted in systematically and significantly overestimating the C_p_(sol,298) values, indicating that their corresponding atom groups in Table 4, (i.e., entry 73, representing the carboxylic acid, and entries 122 and 125, representing the alkyl- and dialkyl-amino groups, respectively) are not applicable in the heat-capacity evaluation of amino acids. These results, however, should not be interpreted as a confirmation of the zwitterionic form of the amino acids as solids, because the basis for the parameters representing the neutral alkyl- and dialkyl-amino groups is at present too small, and a recalibration of the group parameters of Table 4 by applying the non-ionic instead of the zwitter-ionic forms could well lead to better-conforming GA-based results of the non-ionic forms with the experimental data.

In contrast, analogous comparative calculations based on the V_m_ method [1] revealed only minor prediction differences between the ionic and the non-ionic forms, which was to be expected as the "true" molecular volumes of both the prototropic forms are very similar. Typical examples listed in Table 10 demonstrate these observations. The MAPD between the experimental data and those calculated for the zwitterionic forms in Table 10 was 3.36 J/mol/K on applying the GA method and 4.02 J/mol/K when using the V_m_ approach.

As a consequence, and despite their excellent predictive quality, both the GA- and the V_m_-based methods are not suitable to answer the question as to which form the amino acids exist as solids.

In the optimization process for the evaluation of the atom and special group contributions of Table 7, it turned out that 51 compounds had to be eliminated as outliers and have been collected in a separate list, available in the Appendix A. This list again largely corresponded to the one resulting from the V_m_ optimization procedure. The remaining 800 compounds finally supported 126 atom and special groups valid for C_p_(sol,298) predictions (row A in Table 7). Despite the smaller number of valid groups as compared to that of Table 3 for the liquid heat capacities, with 65% they cover an even slightly larger percentage of ChemBrain’s representative database.

## 5. Conclusions

The present paper is extending the series of publications [6,7,8,9] about the direct and indirect calculation of 14 molecular properties (enthalpy of combustion, formation, vaporization, sublimation and solvation, entropy of fusion, logP_o/w_, logS, logγ_inf_, refractivity, polarizability, toxicity, liquid viscosity, and surface tension) by means of a single computer algorithm, adding two further molecular properties, the heat capacity for the liquid and solid phase of molecules. A comparison of the prediction quality of the present GA method for the heat capacities with that based on the "true" molecular volume [1] published recently proved a significantly higher accuracy over the latter. This was accomplished by directly addressing the deficiencies of the molecular volume approach, particularly its inevitable neglect of the intermolecular formation of hydrogen bridges of the OH groups as well as its non-consideration of the cyclization effect of saturated rings over ring-open forms on the heat capacity of both the liquid and solid phases. However, since the group additivity method in principle lacks the comprehensive range of the molecular volume approach, both prediction methods are beneficial in their own right—and they complement each other all the more, as in most cases they confirm each other’s result within explicable deviations. Therefore, in the present ongoing project ChemBrain IXL, version 5.9, available from Neuronix Software (www.neuronix.ch, Rudolf Naef, Lupsingen, Switzerland), the results of both methods are added to the database, the group-additivity result carrying the suffix "calc" and the volume-derived one the suffix "pred".

## Figures and Tables

**Figure 1 molecules-25-01147-f001:**
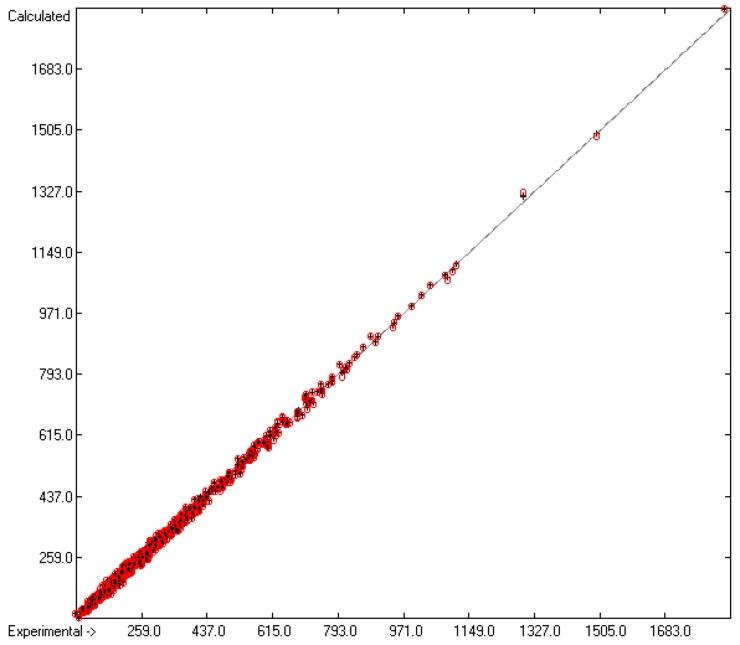
Correlation diagram of the C_p_(liq,298) data (in J/mol/K). The cross-validation data are superimposed as red circles. (*n* = 1111; *r*^2^ = 0.998; *q*^2^ = 0.9975; regression line: intercept = 0.7993; slope = 0.9977).

**Figure 2 molecules-25-01147-f002:**
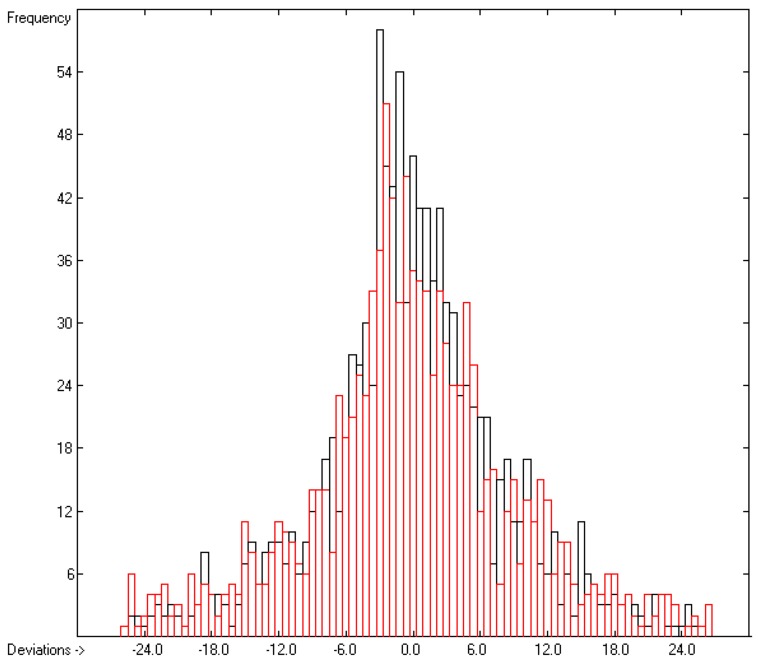
Histogram of the liquid heat-capacity data. The deviations are in J/mol/K. The cross-validation data are superimposed as red bars. (S = ± 9.19; Experimental value range: 81.92–1849 J/molK).

**Figure 3 molecules-25-01147-f003:**
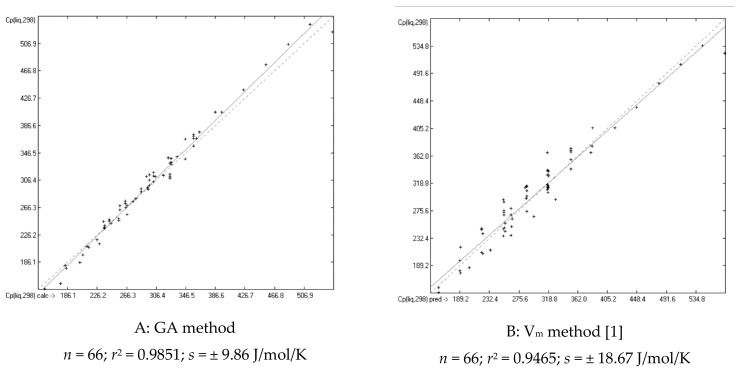
Correlation diagrams of the calculated vs. experimental C_p_(liq,298) data of saturated alcohols (in J/mol/K) based on A: the present GA; B: the V_m_ method.

**Figure 4 molecules-25-01147-f004:**
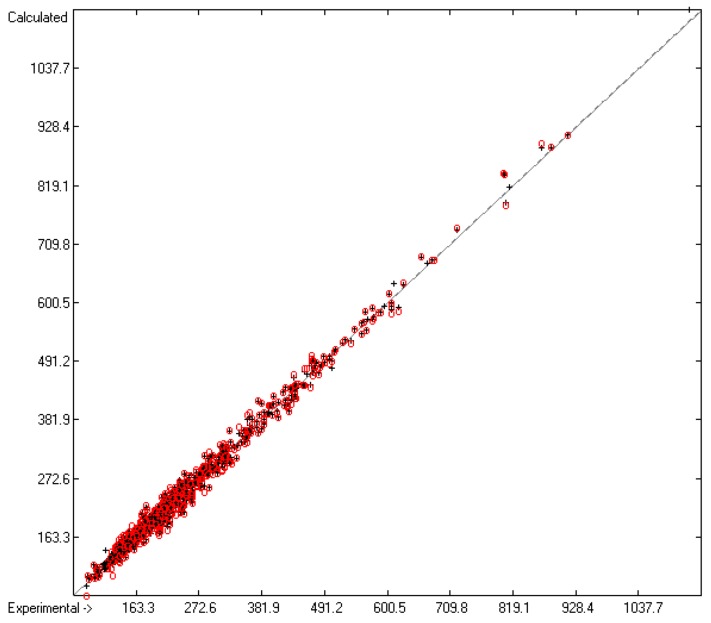
Correlation diagram of the C_p_(sol,298) data (in J/mol/K). The cross-validation data are superimposed as red circles. (*n* = 734; *r*^2^ = 0.9915; *q*^2^ = 0.9874; regression line: intercept = −0.0999; slope = 0.9984).

**Figure 5 molecules-25-01147-f005:**
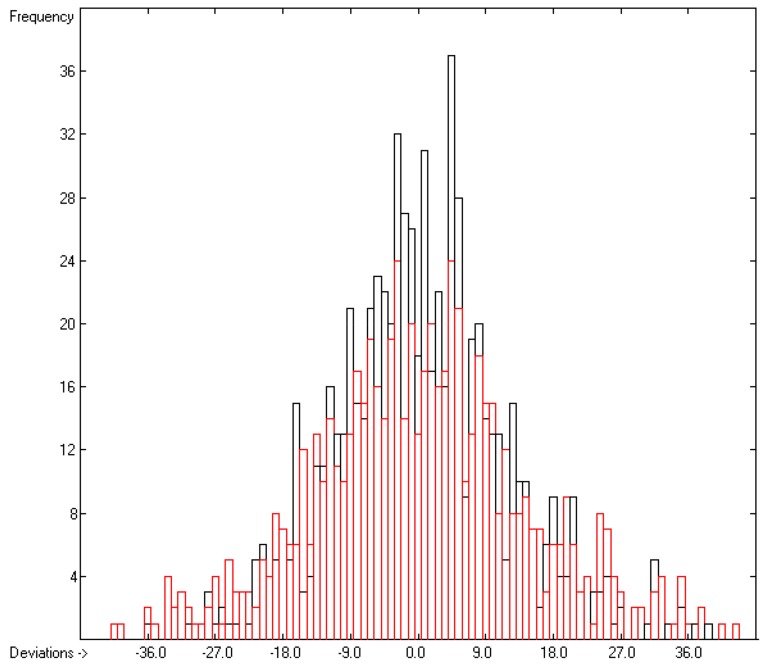
Histogram of the solid heat-capacity data. The deviations are in J/mol/K. The cross-validation data are superimposed as red bars. (*s* = ±14.23; Experimental value range: 78.7–1129 J/molK).

**Table 1 molecules-25-01147-t001:** Atom-group examples for ionic liquids and their meaning.

Atom Type	Neighbours	Meaning^a^	Example
B(-)	F4	**B**F_4_^-^	tetrafluoroborate
C sp3	H2CN(+)	C**C**H_2_N(+)	C1 in tetraalkylammonium
C sp3	H2CP(+)	C**C**H_2_P(+)	C1 in tetraalkylphosphonium
C sp3	H2CS(+)	C**C**H_2_S(+)	C1 in trialkylsulfonium
C(-) sp3	C3	C_3_**C**^-^	central C^-^ in tricyanocarbeniate
C aromatic	H:C:N(+)	C:**C**H:N^+^	C2 in pyridinium
C(+) aromatic	H:N2	N:**C**^+^H:N	C2 in imidazolium
C sp	B#N(-)	B^-^(**C**#N)	C in tetracyanoborate
C sp	C#N(-)	C^-^(**C**#N)	cyano-C in tricyanocarbeniate
C sp	N#N(-)	N^-^(**C**#N)	C in dicyanoamide
C sp	=N=S(-)	N=**C**=S^-^	thiocyanate
N(+) sp3	C4	**N**^+^C_4_	tetraalkylammonium
N(+) sp2	O2=O(-)	**N**O_3_^-^	nitrate
N aromatic	C2:C(+)	C-**N**(C):C^+^	N1 and N3 in 1,3-dialkylimidazolium
N(+) aromatic	C:C2	C:**N**^+^(C):C	N in 1-alkylpyridinium
N(-)	C2	C-**N**^-^-C	N^-^ in dicyanoamide
N(-)	CS	C-**N**^-^-S	N^-^ in saccharinate
N(-)	S2	S-**N**^-^-S	bis(trifluoromethanesulfonyl)amide
P4	CO2=O(-)	C**P**O3^-^	alkylphosphonate
P(+)	C4	**P**C_4_^+^	tetraalkylphosphonium
P(-)	C3F3	F_3_**P**^-^C_3_	tris(pentafluoroethyl)trifluorophosphate
P(-)	F6	**P**F_6_^-^	hexafluorophosphate
S(+)	C3	C_3_**S**^+^	trialkylsulfonium
S4	CN=O2(-)	C**S**(O_2_)N^-^	bis(trifluoromethanesulfonyl)amide
S4	CO=O2(-)	C**S**O_3_^-^	alkylsulfonate
S4	O2=O2(-)	**S**O_4_^-^	alkylsulfate

^a^ The central atom defined by the atom type is indicated by a bold character.

**Table 2 molecules-25-01147-t002:** Refined atom and special groups and their meaning.

Atom Type	Neighbours	Meaning
O(prim)	HC	Primary alcohol
O(sec)	HC	Secondary alcohol
O(tert)	HC	Tertiary alcohol
Endocyclic bonds	No of single bonds	Count single bonds in cyclic ring
Angle60		Bond angle < 60 deg
Angle 90		Bond angle between 60 and 90 deg
Angle102		Bond angle between 90 and 102 deg
(COH)n	n > 1	Molecule contains more than 1 OH group
H	H Acceptor	Intramolecular H bridge between acidic H (on O, N or S) and basic acceptor (O, N or F)

**Table 3 molecules-25-01147-t003:** Atom groups and their contributions for the heat-capacity calculation of liquids.

Entry	Atom Type	Neighbours	Contribution	Occurrences	Molecules
1	B	C3	240	1	1
2	B(-)	C4	698.66	2	2
3	B(-)	F4	51.21	6	6
4	C sp3	H3C	37.03	1555	790
5	C sp3	H3N	100.02	127	101
6	C sp3	H3N(+)	147.91	20	18
7	C sp3	H3O	81.29	84	66
8	C sp3	H3S	84.43	17	13
9	C sp3	H3S(+)	172.17	1	1
10	C sp3	H3P	217.59	1	1
11	C sp3	H3Si	71	71	18
12	C sp3	H2BC	–37.03	3	1
13	C sp3	H2C2	30.06	3249	696
14	C sp3	H2CN	90.52	222	146
15	C sp3	H2CN(+)	142.85	78	52
16	C sp3	H2CO	73.86	477	243
17	C sp3	H2CS	75.25	38	27
18	C sp3	H2CS(+)	136.25	29	10
19	C sp3	H2CP	252.08	2	1
20	C sp3	H2CP(+)	71.8	12	3
21	C sp3	H2CCl	63.67	38	30
22	C sp3	H2CBr	63.93	26	21
23	C sp3	H2CJ	67.73	10	9
24	C sp3	H2CSi	60.71	18	8
25	C sp3	H2N2	151.55	4	2
26	C sp3	H2NO	157.51	12	12
27	C sp3	H2O2	111.47	4	4
28	C sp3	H2S2	128.41	1	1
29	C sp3	HC3	21.11	303	196
30	C sp3	HC2N	81.86	14	14
31	C sp3	HC2N(+)	159.6	3	3
32	C sp3	HC2O	67.45	107	87
33	C sp3	HC2S	67.46	10	9
34	C sp3	HC2Si	36.48	1	1
35	C sp3	HC2Cl	56.56	9	9
36	C sp3	HC2Br	56.84	4	4
37	C sp3	HC2J	62.21	2	2
38	C sp3	HCNO(+)	176.8	3	1
39	C sp3	HCO2	96.7	3	3
40	C sp3	HCF2	157.26	1	1
41	C sp3	HCFCl	73.89	1	1
42	C sp3	HCCl2	86.15	9	8
43	C sp3	HCClBr	89.68	1	1
44	C sp3	HCBr2	82.85	2	1
45	C sp3	C4	7.78	62	51
46	C sp3	C3N	81.33	5	4
47	C sp3	C3N(+)	55.23	3	3
48	C sp3	C3O	57.44	23	21
49	C sp3	C3S	57.43	7	5
50	C sp3	C3F	43.66	5	3
51	C sp3	C3Cl	50.92	1	1
52	C sp3	C3Br	54.62	1	1
53	C sp3	C2N2(+)	223.66	2	2
54	C sp3	C2O2	99.71	1	1
55	C sp3	C2F2	50.88	78	13
56	C sp3	C2FCl	64.31	5	2
57	C sp3	C2Cl2	87.42	2	2
58	C sp3	CNF2	112.99	3	1
59	C sp3	CF3	66.92	31	23
60	C sp3	CSF2	0	1	1
61	C sp3	CPF2(-)	44.23	6	2
62	C sp3	CF2Cl	89.67	4	4
63	C sp3	CF2Br	86.41	7	4
64	C sp3	CFCl2	88.11	3	2
65	C sp3	CCl3	102.55	8	8
66	C sp3	SF3	102.78	151	78
67	C(-) sp3	C3	131.76	1	1
68	C sp2	H2=C	35.64	61	59
69	C sp2	HC=C	22.79	195	107
70	C sp2	HC=N	95.94	4	4
71	C sp2	HC=O	54.97	23	23
72	C sp2	H=CN	92.25	166	87
73	C sp2	H=CO	42.8	11	10
74	C sp2	H=CS	87.41	5	5
75	C sp2	H=CCl	56.41	5	3
76	C sp2	H=CSi	43.69	4	4
77	C sp2	HN=N	32.91	3	3
78	C sp2	HN=O	100.63	3	3
79	C sp2	HO=O	58.82	7	7
80	C sp2	H=NS	15.48	2	2
81	C sp2	C2=C	16.22	54	44
82	C sp2	C2=N	333.48	1	1
83	C sp2	C=CN	89.61	3	2
84	C sp2	C2=O	50.28	49	49
85	C sp2	C=CO	36.16	5	5
86	C sp2	C=CS	74.31	5	4
87	C sp2	C=CCl	160.28	1	1
88	C sp2	CN=O	87.09	12	12
89	C sp2	CN=O(-)	87.78	1	1
90	C sp2	C=NS	8.06	1	1
91	C sp2	CO=O	43.25	216	158
92	C sp2	CO=O(-)	27.63	8	7
93	C sp2	C=OS	0	1	1
94	C sp2	C=OCl	70.53	7	6
95	C sp2	=CF2	56.5	2	1
96	C sp2	=CCl2	77.26	5	4
97	C sp2	N2=N	56.88	1	1
98	C sp2	N2=O	131.83	3	3
99	C sp2	NO=O	98.32	1	1
100	C sp2	O2=O	50.04	5	5
101	C aromatic	H:C2	22	1115	238
102	C aromatic	H:C:N	42.32	19	13
103	C aromatic	H:C:N(+)	–9.45	53	32
104	C aromatic	H:N2	0		0
105	C aromatic	:C3	9.57	19	11
106	C aromatic	C:C2	11.58	251	152
107	C aromatic	C:C:N	30.59	8	7
108	C aromatic	C:C:N(+)	–2.69	11	11
109	C aromatic	:C2N	71.34	31	29
110	C aromatic	:C2N(+)	118.05	11	8
111	C aromatic	:C2:N	31.72	3	3
112	C aromatic	:C2O	33.82	46	28
113	C aromatic	:C2S	88.68	7	7
114	C aromatic	:C2Si	37.2	10	7
115	C aromatic	:C2F	37.06	54	17
116	C aromatic	:C2Cl	41.28	19	15
117	C aromatic	:C2Br	52.57	11	8
118	C aromatic	:C2J	43.43	3	3
119	C(+) aromatic	H:N2	–155.06	74	74
120	C sp	B#N(-)	–130.93	8	2
121	C sp	H#C	38.78	6	5
122	C sp	C#C	23.99	10	7
123	C sp	=C2	25.16	4	4
124	C sp	C#N	48.78	35	31
125	C sp	C#N(-)	–9.81	3	1
126	C sp	#CSi	49.58	2	1
127	C sp	N#N(-)	–2.88	12	6
128	C sp	=N2	–89.69	1	1
129	C sp	=N=O	–20.75	8	5
130	C sp	=N=S(-)	43.63	3	3
131	N sp3	H2C	–4.35	33	28
132	N sp3	H2C(pi)	0.31	9	9
133	N sp3	H2N	48.67	5	4
134	N sp3	HC2	–71.79	21	20
135	N sp3	HC2(pi)	–72.44	14	14
136	N sp3	HC2(2pi)	–103.28	6	6
137	N sp3	HCN	–15.74	4	3
138	N sp3	HCN(pi)	–13	1	1
139	N sp3	HCS(pi)	–21.52	1	1
140	N sp3	C3	–160.23	33	28
141	N sp3	C3(pi)	–149.65	17	14
142	N sp3	C3(2pi)	–180.02	3	3
143	N sp3	C3(3pi)	–165.46	1	1
144	N sp3	C2N	–91.6	2	2
145	N sp3	C2N(2pi)	–143.37	2	2
146	N sp3	C2N(3pi)	–160.71	1	1
147	N sp2	H=C	–243.13	1	1
148	N sp2	C=C	15.82	17	13
149	N sp2	C=N	–20.24	2	1
150	N sp2	C=N(+)	–42.22	1	1
151	N sp2	=CN	0	3	3
152	N sp2	=CO	–53.43	1	1
153	N aromatic	C2:C(+)	–0.31	148	74
154	N aromatic	:C2	−16.75	15	15
155	N(+) sp3	H3C	−44.33	1	1
156	N(+) sp3	H2C2	−140.41	4	4
157	N(+) sp3	HC3	−291.64	1	1
158	N(+) sp3	C4	−372.93	13	13
159	N(+) sp2	C=NO(-)	0	1	1
160	N(+) sp2	CO=O(-)	−45.71	25	17
161	N(+) sp2	O2=O(-)	5.99	4	4
162	N(+) aromatic	C:C2	14.22	32	32
163	N(-)	C2	62.36	6	6
164	N(-)	CS	−32.57	1	1
165	N(-)	S2	33.36	73	73
166	O(prim)	HC	14.35	102	89
167	O(sec)	HC	36.17	47	47
168	O(tert)	HC	58	11	11
169	O	HC(pi)	48.39	57	46
170	O	HP	−119.34	1	1
171	O	HS	39.11	1	1
172	O	C2	−59.32	170	98
173	O	C2(pi)	−26.57	191	149
174	O	C2(2pi)	−15.47	22	12
175	O	CN(+)(pi)	55.55	3	3
176	O	CN(2pi)	0	1	1
177	O	CS	16.03	8	8
178	O	CP(pi)	22.25	3	1
179	O	CSi	−22.04	20	5
180	O	Si2	−21.83	19	7
181	P4	C2O=O(-)	−344.96	1	1
182	P4	CO2=O(-)	0	1	1
183	P4	O3=O	0	1	1
184	P(+)	C4	−95.06	3	3
185	P(-)	C3F3	33.12	2	2
186	P(-)	F6	96.53	9	9
187	S2	HC	0.94	19	19
188	S2	HC(pi)	−25.44	1	1
189	S2	C2	−53.94	19	19
190	S2	C2(pi)	−77.07	2	2
191	S2	C2(2pi)	−89.86	7	7
192	S2	CS	−11.11	8	4
193	S4	C2=O	−23.45	2	2
194	S4	C2=O2	−18.86	1	1
195	S4	CN=O2	0	1	1
196	S4	CN=O2(-)	5.69	147	74
197	S4	CO=O2(-)	4.61	9	9
198	S4	O2=O2(-)	0	9	9
199	S(+)	C3	−203.28	10	10
200	Si	C4	−89.09	11	10
201	Si	C3O	−49.41	6	3
202	Si	C3Cl	−25.49	1	1
203	Si	C2O2	9.5	16	6
204	Si	C2Cl2	25.44	3	3
205	Si	CCl3	86.83	3	3
206	Si	O4	0	5	5
207	(COH)n	n > 1	−27.73	20	19
208	H	H Acceptor	−21.43	3	3
209	Endocyclic bonds	No of single bds	−3.92	1341	243
210	Angle60		4.13	69	19
211	Angle90		1.7	63	19
A	Based on	Valid groups	134		1176
B	Goodness of fit	R^2^	0.998		1111
C	Deviation	Average	6.09		1111
D	Deviation	Standard	8.24		1111
E	K-fold cv	K	10		1060
F	Goodness of fit	Q^2^	0.9975		1060
G	Deviation	Average (cv)	6.85		1060
H	Deviation	Standard (cv)	9.19		1060

**Table 4 molecules-25-01147-t004:** Experimental C_p_(liq,298) data of 66 alkanols, compared with prediction values calculated by the present the group-additivity (GA) and the V_m_ [1] method (in J/mol/K).

Molecule Name	C_p_(liq,298) calc. (GA)	Dev. (%)	C_p_(liq,298) exp.	Dev. (%)	C_p_(liq,298) calc. (V_m_)
1-Propanol	155.30	−5.73	146.88	−8.12	158.80
2-Propanol	177.70	−15.07	154.43	−2.83	158.80
2-Methyl-1-propanol	183.40	−1.30	181.05	−5.05	190.20
1-Butanol	185.40	−4.65	177.16	−7.81	191.00
Cyclopentanol	204.20	−10.14	185.40	−9.76	203.50
2-Butanol	207.70	−5.61	196.67	3.34	190.10
Isopentyl alcohol	213.50	−1.86	209.60	−5.92	222.00
1-Pentanol	215.40	−3.49	208.14	−7.19	223.10
2-Methyl-2-propanol	226.50	−3.61	218.60	12.76	190.70
Cyclohexanol	230.40	−7.97	213.40	−9.89	234.50
3-Methyl-2-butanol	235.80	4.11	245.90	9.68	222.10
Cyclohexanemethanol	236.10	0.17	236.50	−12.05	265.00
3,3-Dimethyl-1-butanol	237.10	−0.43	236.08	−7.51	253.80
3-Pentanol	237.80	0.79	239.70	6.76	223.50
2-Ethyl-1-butanol	243.50	1.28	246.65	−2.74	253.40
2-Methyl-1-pentanol	243.50	1.97	248.40	−2.62	254.90
1-Hexanol	245.50	−1.15	242.70	−5.44	255.90
Cycloheptanol	256.50	−2.51	250.22	−6.03	265.30
2-Methyl-2-butanol	256.60	−3.76	247.30	10.31	221.80
trans-2-Methylcyclohexanol	258.50	1.70	262.98	−1.30	266.40
cis-2-Methylcyclohexanol	258.50	3.89	268.95	1.51	264.90
4-Methyl-2-pentanol	265.90	2.36	272.34	6.66	254.20
3-Methyl-2-pentanol	265.90	3.62	275.89	7.93	254.00
Cyclohexaneethanol	266.20	−0.08	266.00	−12.03	298.00
2-Hexanol	267.90	−4.52	256.31	0.20	255.80
3-Hexanol	267.90	0.51	269.27	5.49	254.50
1-Heptanol	275.50	−0.25	274.81	−4.84	288.10
1-Methylcyclohexanol	279.20	−0.05	279.05	5.03	265.00
2-Methyl-2-pentanol	286.70	0.81	289.03	11.84	254.80
3-Methyl-3-pentanol	286.70	2.25	293.30	13.57	253.50
2,4-Dimethyl-3-pentanol	294.00	5.77	312.00	8.53	285.40
5-Methyl-2-hexanol	296.00	−0.27	295.20	2.74	287.10
Cyclohexanepropanol	296.20	−1.09	293.00	−12.70	330.20
2-Heptanol	297.90	0.24	298.63	3.59	287.90
3-Heptanol	297.90	5.19	314.20	8.37	287.90
4-Heptanol	297.90	2.89	306.77	6.15	287.90
2-Ethyl-1-hexanol	303.60	4.38	317.50	−0.28	318.40
2-Methyl-1-heptanol	303.60	3.00	313.00	−2.01	319.30
5-Methyl-1-heptanol	303.60	0.20	304.20	−4.73	318.60
1-Octanol	305.60	2.08	312.10	−2.40	319.60
2-Methyl-2-hexanol	316.70	−1.01	313.54	8.50	286.90
2,5-Dimethyl-3-hexanol	324.00	4.54	339.40	6.25	318.20
2-Methyl-4-heptanol	326.00	1.75	331.80	4.01	318.50
4-Methyl-2-heptanol	326.00	−4.32	312.50	−1.70	317.80
4-Methyl-3-heptanol	326.00	−5.43	309.20	−2.81	317.90
6-Methyl-2-heptanol	326.00	−3.46	315.10	−1.30	319.20
6-Methyl-3-heptanol	326.00	−4.99	310.50	−2.77	319.10
2-Octanol	328.00	0.64	330.10	3.00	320.20
3-Octanol	328.00	3.10	338.50	5.44	320.10
4-Octanol	328.00	1.23	332.09	3.64	320.00
1-Nonanol	335.70	1.55	341.00	−3.37	352.50
2-Methyl-2-heptanol	346.80	−2.73	337.60	5.45	319.20
4-Methyl-4-heptanol	346.80	5.48	366.90	13.25	318.30
2-Nonanol	358.00	−0.47	356.32	1.10	352.40
3-Nonanol	358.00	4.18	373.63	5.71	352.30
4-Nonanol	358.00	2.68	367.86	4.26	352.20
5-Nonanol	358.00	3.44	370.75	4.98	352.30
3,7-Dimethyl-1-octanol	361.80	1.47	367.21	−4.05	382.10
n-Decyl alcohol	365.70	3.00	377.00	−1.83	383.90
5-Decanol	388.10	4.35	405.77	5.24	384.50
1-Undecanol	395.80	2.59	406.34	−2.60	416.90
1-Dodecanol	425.80	2.88	438.42	−2.44	449.10
1-Tridecanol	455.90	4.22	476.00	−1.11	481.30
Myristyl alcohol	486.00	3.91	505.80	−1.52	513.50
1-Pentadecanol	516.00	3.57	535.10	−1.98	545.70
1-Hexadecanol	546.10	−4.26	523.80	−10.33	577.90
MAPD		3.02		5.51	

**Table 5 molecules-25-01147-t005:** Experimental C_p_(liq,298) data of four linear alkanes and four related cycloalkanes, compared with the prediction values calculated by the present GA and the V_m_ [1] method (in J/mol/K).

Molecule Name	C_p_(liq,298) calc. (GA)	Dev. (%)	C_p_(liq,298) Exp.	Dev. (%)	C_p_(liq,298) Calc.(V_m_)
Cyclopentane	130.70	−1.48	128.80	−15.30	148.50
Pentane	164.30	1.73	167.19	−0.01	167.20
Cyclohexane	156.80	0.82	158.10	−12.14	177.30
Hexane	194.30	1.70	197.66	0.08	197.50
Cycloheptane	183.00	−1.32	180.61	−13.23	204.50
Heptane	224.40	0.41	225.33	−1.10	227.80
Cyclooctane	209.10	2.98	215.53	−10.29	237.70
Octane	254.40	0.50	255.68	−0.99	258.20

**Table 6 molecules-25-01147-t006:** Experimental C_p_(liq,298) data of 122 ionic liquids, compared with prediction values calculated by the present GA and the V_m_ [1] method (in J/mol/K).

Molecule Name	C_p_(liq,298) GA-Calc.	Dev. (%)	C_p_(liq,298) Exp.	Dev. (%)	C_p_(liq,298) V_m_-Calc.
1-Ethyl-3-methylimidazolium bromide	256.40	3.17	264.80	3.29	256.10
1-Propyl-3-methylimidazolium bromide	286.50	−1.81	281.40	−1.92	286.80
1-Ethyl-3-methylimidazolium thiocyanate	300.00	−6.59	281.45	−4.96	295.40
1-Ethyl-3-methylimidazolium acetate	321.10	0.25	321.90	7.86	296.60
1-Ethyl-3-methylimidazolium tetrafluoroborate	307.60	0.16	308.10	1.95	302.10
1,3-Dimethylimidazolium methosulfate	326.20	4.34	341.00	10.50	305.20
1-Butyl-3-methylimidazolium chloride	316.50	0.16	317.00	3.63	305.50
1-Ethyl-3-methylimidazolium dicyanamide	313.00	0.52	314.64	−0.08	314.90
1-Butyl-3-methylimidazolium bromide	316.50	0.06	316.70	−0.38	317.90
*N*-Methyl-2-hydroxyethylammonium propionate	333.30	−1.62	328.00	2.10	321.10
1-Ethyl-3-methylimidazolium methanesulfonate	345.40	0.03	345.50	6.80	322.00
1-Ethyltetrahydrothiophenium dicyanamide	339.60	−1.26	335.38	1.34	330.90
1-Ethyl-3-methylimidazolium methylsulfate	353.70	−3.72	341.00	0.67	338.70
1-Ethyl-3-methylimidazolium hexafluorophosphate	353.00	−2.74	343.60	0.67	341.30
1-Butyl-3-methylimidazolium iodide	316.50	−0.80	314.00	−8.82	341.70
1-Benzyl-3-methylimidazolium chloride	341.00	−0.47	339.40	−1.30	343.80
1-Ethylpyridinium triflate	348.70	0.88	351.80	0.71	349.30
1-Ethyl-3-methylimidazolium trifluoromethylsulfonate	363.80	−0.28	362.80	3.64	349.60
*N*-Methyl-2-hydroxyethylammonium butanoate	363.40	−0.66	361.00	2.13	353.30
1-Butyl-3-methylimidazolium thiocyanate	360.20	6.44	385.00	7.53	356.00
1-Butyl-3-methylimidazolium acetate	381.20	0.52	383.20	6.16	359.60
1-Butyl-3-methylimidazolium tetrafluoroborate	367.70	−0.79	364.80	0.63	362.50
1-Ethyl-3-methylimidazolium ethosulfate	383.30	−1.40	378.00	2.99	366.70
1-Propyl-3-methylimidazolium hexafluorophosphate	383.00	−2.30	374.40	0.64	372.00
1-Butyl-3-methylimidazolium dicyanoamide	373.10	−2.22	365.00	−2.19	373.00
1-Butyl-3-methylimidazolium trifluoroacetate	411.10	−0.71	408.20	5.66	385.10
*N*-Methyl-2-hydroxyethylammonium pentanoate	393.40	1.90	401.00	3.87	385.50
1-Butyltetrahydrothiophenium dicyanamide	399.70	−1.14	395.19	0.96	391.40
1-Butyl-3-methylpyridinium tetrafluoroborate	379.20	2.27	388.00	−1.26	392.90
1-Ethyl-3-methylpyridinium ethylsulfate	394.70	−1.47	389.00	−2.08	397.10
1-Butyl-3-methylimidazolium methosulfate	413.80	0.53	416.00	4.50	397.30
1-Benzyl-3-methylimidazolium tetrafluoroborate	392.20	−1.21	387.50	−3.23	400.00
1-Butyl-3-methylimidazolium hexafluorophosphate	413.10	−1.32	407.70	1.37	402.10
1-Butyl-1-methylpyrrolidinium dicyanamide	397.80	3.68	413.00	1.86	405.30
1-Butyl-3-methylimidazolium trifluoromethylsulfonate	423.90	−1.65	417.00	1.75	409.70
1-Hexyl-3-methylimidazolium tetrafluoroborate	427.90	−2.86	416.00	−1.32	421.50
1-Pentyl-3-methylimidazolium hexafluorophosphate	443.10	−1.30	437.40	1.03	432.90
1-Butyl-1-methylpyrrolidinium trifluoromethanesulfonate	448.70	−3.15	435.00	−0.55	437.40
1-Ethyl-3-methylimidazolium toluenesulfonate	486.30	−0.43	484.20	6.84	451.10
1-Hexyl-3-methylimidazolium trifluoromethylsulfonate	484.00	3.64	502.30	6.43	470.00
1-Octyl-3-methylimidazolium tetrafluoroborate	488.00	2.01	498.00	2.99	483.10
1-Ethyl-3-methylimidazolium bis(trifluoromethanesulfonyl) amide	506.70	−1.34	500.00	3.36	483.20
N-Ethylpyridinium bis(trifluoromethylsulfonyl)amide	491.50	2.12	502.15	2.98	487.20
1-Ethyl-3-methylimidazolium 2-(2-methoxyethoxy)ethylsulfate	530.50	−0.86	526.00	6.29	492.90
1-Heptyl-3-methylimidazolium hexafluorophosphate	503.30	−0.54	500.60	1.34	493.90
1-Isopropyl-3-methylimidazolium bis(trifluoromethanesulfonyl) amide	535.20	−1.00	529.90	3.25	512.70
1-Propyl-3-methylimidazolium bis(trifluoromethanesulfonyl) amide	536.70	−0.34	534.90	4.08	513.10
1-Butyl-3-methylimidazolium toluenesulfonate	546.40	0.36	548.40	6.20	514.40
*N*-Ethyl-2-methylpyridinium bis(trifluoromethylsulfonyl)amide	535.30	−0.15	534.50	3.55	515.50
1-Octyl-3-methylimidazolium hexafluorophosphate	533.30	0.52	536.10	2.16	524.50
1-Cyclopropylmethyl-3-methylimidazolium bis(trifluoromethanesulfonyl) amide	551.50	−2.30	539.10	1.21	532.60
Trimethyl butylammonium bis(trifluoromethylsulfonyl)amide	561.10	−0.34	559.20	4.02	536.70
1,2-Diethylpyridinium bis(trifluoromethanesulfonyl) amide	565.40	0.12	566.10	4.82	538.80
1-Butyl-3-methylimidazolium bis(trifluoromethanesulfonyl) amide	566.80	−0.16	565.90	4.22	542.00
1-sec-Butyl-3-methylimidazolium bis(trifluoromethanesulfonyl) amide	565.30	−1.47	557.10	2.64	542.40
1-Methyl-1-propylpyrrolidinium bis(trifluoromethanesulfonyl) amide	561.50	−1.35	554.00	1.93	543.30
1-Isobutyl-3-methylimidazolium bis(trifluoromethanesulfonyl) amide	564.80	−1.38	557.10	2.48	543.30
*N*-Propyl-2-methylpyridinium bis(trifluoromethylsulfonyl)amide	565.40	−1.33	557.96	2.18	545.80
*N*-Butylpyridinium bis(trifluoromethanesulfonyl) amide	551.60	2.63	566.52	3.50	546.70
1-Nonyl-3-methylimidazolium hexafluorophosphate	563.40	1.05	569.40	2.79	553.50
*N*-Octylisoquinolinium thiocyanate	528.30	−1.21	522.00	−6.88	557.90
1-Butyltetrahydrothiophenium bis(trifluoromethylsulfonyl) amide	593.40	0.44	596.00	6.12	559.50
*N*-Ethyl-4-dimethylaminopyridinium bis(trifluoromethanesulfonyl) amide	591.20	0.52	594.30	4.37	568.30
1-Pentyl-3-methylimidazolium bis(trifluoromethanesulfonyl) amide	596.90	−0.22	595.60	4.08	571.30
1-Ethyl-2-propylpyridinium bis(trifluoromethanesulfonyl) amide	595.40	−0.25	593.90	3.25	574.60
1-Isobutyl-1-methylpyrrolidinium bis(trifluoromethylsulfonyl)amide	589.60	−1.27	582.20	1.03	576.20
1-Isobutyl-3-methylpyridinium bis(trifluoromethylsulfonyl)amide	576.30	0.47	579.00	0.40	576.70
*N*-Butyl-3-methylpyridinium bis(trifluoromethylsulfonyl)amide	578.20	−0.02	578.10	0.14	577.30
1-Butyl-1-methylpyrrolidinium bis(trifluoromethylsulfonyl)amide	591.50	−3.41	572.00	−1.05	578.00
1-Butyl-3-cyanopyridinium bis(trifluoromethylsulfonyl)amide	590.00	−0.68	586.00	1.11	579.50
1-Benzyl-3-methylimidazolium bis(trifluoromethylsulfonyl) amide	591.20	2.73	607.80	4.34	581.40
1-Decyl-3-methylimidazolium hexafluorophosphate	593.40	1.64	603.30	3.03	585.00
1-Pentyltetrahydrothiophenium bis(trifluoromethylsulfonyl) amide	623.50	0.56	627.00	6.19	588.20
1-Hexyl-3-methylimidazolium bis(trifluoromethanesulfonyl) amide	626.90	0.37	629.20	4.04	603.80
1-Butyl-1-methylpiperidinium bis(trifluoromethylsulfonyl)amide	617.70	−1.68	607.50	0.23	606.10
1-Ethyl-2-butylpyridinium bis(trifluoromethanesulfonyl) amide	625.50	−0.30	623.60	2.79	606.20
*N*-Hexylpyridinium bis(trifluoromethanesulfonyl) amide	611.80	0.03	612.00	0.83	606.90
1-Methyl-1-pentylpyrrolidinium bis(trifluoromethanesulfonyl) amide	621.60	0.16	622.60	2.28	608.40
1-Butyl-3-methylimidazolium octylsulfate	623.80	1.76	635.00	3.92	610.10
1-Cyclohexylmethyl-3-methylimidazolium bis(trifluoromethanesulfonyl) amide	617.50	−0.06	617.10	0.26	615.50
1-Hexyltetrahydrothiophenium bis(trifluoromethylsulfonyl) amide	653.60	−1.18	646.00	4.30	618.20
*N*-Butyl-4-dimethylaminopyridinium bis(trifluoromethanesulfonyl) amide	651.40	0.96	657.71	4.71	626.70
1-Heptyl-3-methylimidazolium bis(trifluoromethanesulfonyl) amide	657.00	0.33	659.20	4.31	630.80
1-Ethyl-2-pentylpyridinium bis(trifluoromethanesulfonyl) amide	655.60	−0.44	652.70	2.47	636.60
1-Hexyl-3-methylpyridinium bis(trifluoromethylsulfonyl)amide	638.30	−2.29	624.00	−2.20	637.70
1-Hexyl-1-methylpyrrolidinium bis(trifluoromethanesulfonyl) amide	651.70	0.52	655.10	2.49	638.80
1-Hexyl-4-cyanopyridinium bis(trifluoromethylsulfonyl)amide	650.10	−2.70	633.00	−1.04	639.60
1-Hexyl-3-cyanopyridinium bis(trifluoromethylsulfonyl)amide	650.10	1.20	658.00	2.78	639.70
1-Dodecyl-3-methylimidazolium hexafluorophosphate	653.60	1.91	666.30	2.97	646.50
1-Heptyltetrahydrothiophenium bis(trifluoromethylsulfonyl) amide	683.60	0.20	685.00	5.37	648.20
1-Octyl-3-methylimidazolium bis(trifluoromethanesulfonyl) amide	687.10	0.45	690.20	3.80	664.00
1-Methyl-1-heptylpyrrolidinium bis(trifluoromethanesulfonyl) amide	681.70	0.50	685.10	2.96	664.80
1-Octylpyridinium bis(trifluoromethylsulfonyl)amide	671.90	2.06	686.00	2.93	665.90
1-Ethyl-2-hexylpyridinium bis(trifluoromethanesulfonyl) amide	685.60	−0.01	685.50	2.68	667.10
1-Octyltetrahydrothiophenium bis(trifluoromethylsulfonyl) amide	713.70	0.74	719.00	5.63	678.50
1-(3,4,5,6-Perfluorohexyl)-3-methylimidazolium-3-methylimidazolium bis(trifluoromethanesulfonyl) amide	719.30	0.79	725.00	6.40	678.60
4-Dimethylamino-1-hexylpyridinium bis(trifluoromethanesulfonyl) amide	711.50	2.67	731.00	5.99	687.20
1-Ethyl-2-heptylpyridinium bis(trifluoromethanesulfonyl) amide	715.70	0.25	717.50	2.80	697.40
1-Methyl-1-octylpyrrolidinium bis(trifluoromethanesulfonyl) amide	711.80	0.63	716.30	2.36	699.40
1-Octyl-3-cyanopyridinium bis(trifluoromethylsulfonyl)amide	710.20	−0.17	709.00	1.26	700.10
*N*-Hexyl-3-methyl-4-dimethylaminopyridinium bis(trifluoromethanesulfonyl) amide	738.10	−1.81	725.00	1.94	710.90
1-Nonyltetrahydrothiophenium bis(trifluoromethylsulfonyl) amide	743.70	−0.36	741.00	4.01	711.30
Butyl 1-butylnicotinate bis(trifluoromethylsulfonyl)amide	728.90	−3.10	707.00	−0.91	713.40
1-Decyl-3-methylimidazolium bis(trifluoromethanesulfonyl) amide	747.20	1.01	754.80	4.03	724.40
1-Ethyl-2-octylpyridinium bis(trifluoromethanesulfonyl) amide	745.70	0.49	749.40	3.26	725.00
1-Methyl-3-menthyloxymethylimidazolium bis(trifluoromethylsulfonyl)imide	785.90	−0.58	781.40	4.20	748.60
1-Ethyl-2-nonylpyridinium bis(trifluoromethanesulfonyl) amide	775.80	0.35	778.50	3.48	751.40
1-Methyl-1-decylpyrrolidinium bis(trifluoromethanesulfonyl) amide	771.90	0.90	778.90	2.40	760.20
1-Ethyl-3-menthyloxymethylimidazolium bis(trifluoromethylsulfonyl)imide	813.40	0.61	818.40	4.55	781.20
1-Dodecyl-3-methylimidazolium bis(trifluoromethanesulfonyl) amide	807.30	1.57	820.20	4.04	787.10
1-Ethyl-2-decylpyridinium bis(trifluoromethanesulfonyl) amide	805.80	0.67	811.20	2.79	788.60
1-Propyl-3-menthyloxymethylimidazolium bis(trifluoromethylsulfonyl)imide	843.50	−0.49	839.40	3.71	808.30
1-Butyl-3-menthyloxymethylimidazolium bis(trifluoromethylsulfonyl)imide	873.50	–0.84	866.20	3.20	838.50
1-Pentyl-3-menthyloxymethylimidazolium bis(trifluoromethylsulfonyl)imide	903.60	0.15	905.00	3.55	872.90
1-Hexyl-3-menthyloxymethylimidazolium bis(trifluoromethylsulfonyl)imide	933.70	1.15	944.60	4.74	899.80
1-Heptyl-3-menthyloxymethylimidazolium bis(trifluoromethylsulfonyl)imide	963.70	–0.33	960.50	3.25	929.30
1-Octyl-3-menthyloxymethylimidazolium bis(trifluoromethylsulfonyl)imide	993.80	0.12	995.00	3.59	959.30
1-Nonyl-3-menthyloxymethylimidazolium bis(trifluoromethylsulfonyl)imide	1023.80	−0.02	1023.60	2.83	994.60
1-Decyl-3-menthyloxymethylimidazolium bis(trifluoromethylsulfonyl)imide	1053.90	–0.76	1045.90	2.31	1021.70
1-Undecyl-3-menthyloxymethylimidazolium bis(trifluoromethylsulfonyl)imide	1084.00	0.40	1088.30	3.55	1049.70
1-Dodecyl-3-menthyloxymethylimidazolium bis(trifluoromethylsulfonyl)imide	1114.00	0.35	1117.90	2.85	1086.00
Tetradecyl trihexylphosphonium bis(trifluoromethylsulfonyl)amide	1312.10	−1.02	1298.80	−1.35	1316.30
MAPD		1.13		3.08	

**Table 7 molecules-25-01147-t007:** Atom groups and their contributions for the heat-capacity calculation of solids.

Entry	Atom Type	Neighbours	Contribution	Occurrences	Molecules
1	B(-)	F4	2.99	1	1
2	C sp3	H3C	37.12	569	246
3	C sp3	H3N	101.28	50	34
4	C sp3	H3N(+)	99.59	7	4
5	C sp3	H3O	68.06	51	35
6	C sp3	H3S	45.85	3	3
7	C sp3	H3P	131.12	1	1
8	C sp3	H3Si	59.18	14	5
9	C sp3	H2C2	25.45	1427	249
10	C sp3	H2CN	82.72	89	56
11	C sp3	H2CN(+)	81.35	19	15
12	C sp3	H2CO	64.67	210	108
13	C sp3	H2CS	72.27	22	12
14	C sp3	H2CF	53.68	4	1
15	C sp3	H2CCl	50.12	4	1
16	C sp3	H2CBr	54.73	5	2
17	C sp3	H2CJ	52.94	4	1
18	C sp3	H2CSi	63.12	1	1
19	C sp3	H2N2	134.39	13	3
20	C sp3	H2O2	108.47	12	3
21	C sp3	H2S2	−6.67	3	3
22	C sp3	HC3	11.92	164	73
23	C sp3	HC2N	72.94	28	23
24	C sp3	HC2N(+)	70.74	29	28
25	C sp3	HC2O	51.84	161	63
26	C sp3	HC2S	47.22	4	2
27	C sp3	HC2Si	142.27	1	1
28	C sp3	HCN2	137.04	1	1
29	C sp3	HCNO	119.89	7	5
30	C sp3	HCNS	116.08	2	1
31	C sp3	HCO2	112.62	17	14
32	C sp3	HCF2	246.8	1	1
33	C sp3	HCBr2	68.67	1	1
34	C sp3	C4	−2.17	81	48
35	C sp3	C3N	62.09	11	9
36	C sp3	C3N(+)	19.23	2	2
37	C sp3	C3O	27.59	10	10
38	C sp3	C3Cl	78.13	1	1
39	C sp3	C3Br	44.34	1	1
40	C sp3	C2NO	89.86	1	1
41	C sp3	C2O2	91.75	6	5
42	C sp3	C2S2	43.86	5	2
43	C sp3	CF3	69.89	2	2
44	C sp3	CSF2	0	1	1
45	C sp3	CCl3	92.28	4	3
46	C sp2	H2=C	39.96	6	6
47	C sp2	HC=C	16.95	109	65
48	C sp2	HC=N	99.99	13	13
49	C sp2	HC=O	43.05	14	12
50	C sp2	H=CN	33.94	25	17
51	C sp2	H=CO	41.99	3	3
52	C sp2	H=CS	36.19	7	5
53	C sp2	H=CCl	19.44	1	1
54	C sp2	HN=N	102.56	17	14
55	C sp2	HN=O	28.91	4	3
56	C sp2	H=NO	112.57	1	1
57	C sp2	C2=C	5.77	29	22
58	C sp2	C2=N	95.44	14	10
59	C sp2	C2=N(+)	−9.61	2	2
60	C sp2	C=CN	18.4	16	15
61	C sp2	C2=O	27.68	44	30
62	C sp2	C=CO	25.28	15	13
63	C sp2	C=CS	29.08	5	4
64	C sp2	C=CCl	35.2	6	3
65	C sp2	=CN2	38.93	14	14
66	C sp2	=CN2(+)	78.32	7	7
67	C sp2	CN=N	87.15	19	13
68	C sp2	CN=N(+)	118.23	2	1
69	C sp2	CN=O	37.33	131	92
70	C sp2	=CNO	55.09	1	1
71	C sp2	CN=S	46.51	3	3
72	C sp2	CO=O	51.25	208	155
73	C sp2	CO=O(-)	15.25	41	40
74	C sp2	C=OCl	61.46	2	1
75	C sp2	=CS2	45.53	12	2
76	C sp2	N2=N	113.97	5	3
77	C sp2	N2=O	55.72	43	38
78	C sp2	N=NO	91.56	1	1
79	C sp2	N2=S	67.02	7	7
80	C sp2	N=NS	105.95	7	7
81	C sp2	NO=O	63.43	8	8
82	C sp2	NO=S	64.85	3	3
83	C sp2	=NOS	108.01	1	1
84	C sp2	NS=S	62.46	4	3
85	C sp2	O2=O	58.47	5	5
86	C sp2	OS=S	63.23	1	1
87	C aromatic	H:C2	17.96	3232	437
88	C aromatic	H:C:N	24.16	37	20
89	C aromatic	H:C:N(+)	21.47	2	1
90	C aromatic	H:N2	7.08	3	3
91	C aromatic	:C3	8.04	171	57
92	C aromatic	C:C2	6.56	699	307
93	C aromatic	C:C:N	5.85	13	9
94	C aromatic	:C2N	24.29	172	107
95	C aromatic	:C2N(+)	49.48	57	42
96	C aromatic	:C2:N	19.63	13	7
97	C aromatic	:C2O	29.21	184	113
98	C aromatic	:C2P	15.1	6	2
99	C aromatic	:C2S	27.43	43	28
100	C aromatic	:C2Si	58.51	53	12
101	C aromatic	:C2F	31.08	25	9
102	C aromatic	:C2Cl	34.35	57	25
103	C aromatic	:C2Br	37.48	18	9
104	C aromatic	:C2J	48.49	5	3
105	C aromatic	C:N2	18.81	3	1
106	C aromatic	:CN:N	39.03	7	5
107	C aromatic	:C:NO	53.92	1	1
108	C aromatic	:C:NCl	49.04	3	3
109	C aromatic	N:N2	36.22	4	2
110	C aromatic	:N2O	40.81	3	1
111	C(+) aromatic	H:N2	−39.63	3	3
112	C(+) aromatic	:N3	−20.94	2	2
113	C sp	H#C	103.03	2	1
114	C sp	C#C	14.95	8	3
115	C sp	C#N	39.64	28	20
116	C sp	C#N(+)	62.56	1	1
117	C sp	#CSi	0	2	1
118	C sp	#NO	58.07	2	1
119	C sp	=N=O	110.98	6	3
120	N sp3	H2C	−18.62	5	5
121	N sp3	H2C(pi)	15.49	129	98
122	N sp3	H2N	9.51	4	3
123	N sp3	H2S	45.92	9	9
124	N sp3	HC2	−104.07	6	3
125	N sp3	HC2(pi)	−56.77	78	57
126	N sp3	HC2(2pi)	−9.78	82	61
127	N sp3	HCN(pi)	21.22	7	5
128	N sp3	HCN(2pi)	36.09	7	7
129	N sp3	C3	−159.23	15	10
130	N sp3	C3(pi)	−123.38	10	9
131	N sp3	C3(2pi)	−83.83	27	21
132	N sp3	C3(3pi)	−43.15	12	6
133	N sp3	C2N(pi)	−75.07	3	3
134	N sp3	C2N(+)(pi)	−52.69	7	2
135	N sp3	C2N(2pi)	25.39	3	3
136	N sp3	C2N(+)(2pi)	−41.18	2	2
137	N sp2	C=C	−74.59	54	44
138	N sp2	C=N	2.46	5	3
139	N sp2	=CN	−98.01	7	7
140	N sp2	=CN(+)	−14.76	1	1
141	N sp2	=CO	−44.63	24	11
142	N sp2	N=N	17.43	3	2
143	N aromatic	H2:C(+)	4.65	4	2
144	N aromatic	HC:C(+)	40.44	1	1
145	N aromatic	C2:C(+)	−3.11	7	4
146	N aromatic	:C2	−0.16	50	33
147	N(+) sp3	H3C	−6.7	35	34
148	N(+) sp3	H2C2	−73.79	4	4
149	N(+) sp3	HC3	−156.71	1	1
150	N(+) sp3	C4	−233.93	2	2
151	N(+) sp2	CO=O(-)	9.71	72	49
152	N(+) sp2	=CO2(-)	−7.42	2	2
153	N(+) sp2	NO=O(-)	−2.08	10	5
154	N(+) aromatic	C:C2	0	1	1
155	N(+) sp	C#C(-)	25.34	3	3
156	N(+) sp	#CO(-)	0	1	1
157	O(prim)	HC	−23.36	106	60
158	O(sec)	HC	−16.25	117	50
159	O(tert)	HC	−3.34	6	6
160	O	HC(pi)	4.78	218	157
161	O	HN(pi)	16.33	3	3
162	O	HSi	16.93	8	2
163	O	C2	−70.39	60	29
164	O	C2(pi)	−39.71	125	88
165	O	C2(2pi)	−23.91	47	41
166	O	CN(2pi)	−28.51	5	5
167	O	CSi	−30.4	36	9
168	O	N2(2pi)	1.1	7	4
169	O	N2(+)(2pi)	−2.2	2	2
170	O	Si2	−7.14	39	8
171	P3	C3	−2.25	1	1
172	P4	C3=O	2.25	1	1
173	P4	C=OCl2	0	1	1
174	S2	HC	10.73	1	1
175	S2	HC(pi)	16.07	5	5
176	S2	C2	−15.43	12	8
177	S2	C2(pi)	−30.85	14	6
178	S2	C2(2pi)	−10.04	24	17
179	S2	CS	−20.66	4	2
180	S2	CS(pi)	6.91	6	3
181	S4	C2=O	5.33	2	2
182	S4	C2=O2	15.56	5	5
183	S4	CN=O2	2.75	9	9
184	S4	CO=O2(-)	−118.41	1	1
185	Si	C4	−197.51	3	3
186	Si	C3O	−100.79	4	2
187	Si	C3Cl	−107.38	1	1
188	Si	C2O2	−38.26	11	3
189	Si	CO3	10.05	20	3
190	Si	CCl3	57.89	2	2
191	Si	O4	0	9	9
192	(COH)n	n>1	3.46	145	59
193	H	H Acceptor	1.19	61	43
194	Endocyclic bonds	No of single bds	−1.44	998	149
195	Angle60		0.97	15	2
196	Angle90		0.31	12	6
197	Angle102		2.16	284	83
A	Based on	Valid groups	126		802
B	Goodness of fit	*r* ^2^	0.9915		734
C	Deviation	Average	9.36		734
D	Deviation	Standard	12.21		734
E	K-fold cv	K	10		663
F	Goodness of fit	*q* ^2^	0.9874		663
G	Deviation	Average (cv)	11.1		663
H	Deviation	Standard (cv)	14.23		663

**Table 8 molecules-25-01147-t008:** Experimental C_p_(sol,298) data of 31 alkanols, compared with prediction values calculated by the present GA and the V_m_ [1] method (in J/mol/K).

Molecule Name	C_p_(sol,298) GA-calc	Dev. (%)	C_p_(sol,298) exp.	Dev. (%)	C_p_(sol,298) V_m_−calc. [1]
2-Methyl-2-propanol	135.90	6.99	146.11	18.28	119.40
2,2-Dimethyl-1,3-propanediol	158.00	13.75	183.18	13.42	158.60
Erythritol	164.20	−1.42	161.90	0.99	160.30
cis-1,2-Cyclohexanediol	168.00	−4.74	160.40	−4.68	167.90
trans-1,2-Cyclohexanediol	168.00	−2.94	163.20	−3.00	168.10
Pentaerythritol	174.40	7.43	188.40	3.24	182.30
Hexamethyleneglycol	187.90	1.11	190.00	2.47	185.30
Xylitol	206.90	0.05	207.00	10.14	186.00
Ethriol	191.10	10.62	213.80	7.86	197.00
Inositol	223.60	−2.57	218.00	0.50	216.90
2-Adamantanol	193.30	6.71	207.20	−6.13	219.90
1-Adamantanol	195.60	0.56	196.70	−12.10	220.50
Dulcose	246.00	−3.14	238.50	3.31	230.60
Isoborneol	243.10	6.88	261.06	10.21	234.40
Borneol	243.10	6.88	261.06	9.68	235.80
1,8-Octanediol	238.90	−1.07	236.36	−0.23	236.90
Sorbitol	242.30	−1.38	239.00	0.50	237.80
Menthol	250.70	−0.24	250.10	−0.72	251.90
1,9-Nonanediol	264.30	−2.94	256.74	−2.36	262.80
1,10-Decanediol	289.80	−3.77	279.26	−3.34	288.60
1,11-Undecanediol	315.20	−5.85	297.79	−5.58	314.40
Tri-t-butylmethanol	351.80	−0.34	350.60	6.36	328.30
1,12-Dodecanediol	340.70	−3.17	330.23	−3.05	340.30
1-Tridecanol	358.60	5.13	378.00	8.41	346.20
1,13-Tridecanediol	366.20	0.19	366.88	0.21	366.10
Myristyl alcohol	384.00	1.03	388.00	4.28	371.40
1,14-Tetradecanediol	391.60	−3.16	379.61	−3.24	391.90
1-Pentadecanol	409.50	−2.38	400.00	0.88	396.50
1,15-Pentadecanediol	417.10	−10.50	377.45	−10.66	417.70
1-Hexadecanol	435.00	−3.08	422.00	0.09	421.60
1,16-Hexadecanediol	442.50	−3.83	426.18	−4.06	443.50
MAPD		4.00		5.16	

**Table 9 molecules-25-01147-t009:** Experimental C_p_(sol,298) data of 18 alkanes and cycloalkanes, compared with prediction values calculated by the present GA and the V_m_ [1] method (in J/mol/K).

Molecule Name	C_p_(sol,298) GA-Calc.	Dev. (%)	C_p_(sol,298) exp.	Dev. (%)	C_p_(sol,298) V_m_-Calc. [1]
Nortricyclene	130.70	−1.32	129.00	−9.15	140.80
Norbornane	163.80	−8.48	151.00	0.93	149.60
Bicyclo[2.2.2]octane	163.80	−3.87	157.69	−8.44	171.00
Adamantane	183.30	3.53	190.00	−5.95	201.30
Bicyclo[3.3.3]undecane	236.10	−10.74	213.20	−9.19	232.80
Diamantane	222.10	0.58	223.40	−17.32	262.10
Perhydrophenanthrene	279.60	3.42	289.50	−0.73	291.60
Tri-t-butylmethane	339.50	4.31	354.80	15.30	300.50
Cetane	431.00	2.44	441.80	13.26	383.20
Octadecane	481.90	0.77	485.64	11.70	428.80
Docosane	583.80	−3.58	563.60	7.59	520.80
2,11-Dicyclohexyldodecane	563.40	−1.09	557.30	4.65	531.40
1,1-Dicyclohexyldodecane	565.20	−0.46	562.60	5.44	532.00
Hexacosane	685.60	−3.69	661.20	7.43	612.10
Triacontane	787.40	2.65	808.80	13.03	703.40
Dotriacontane	838.40	−4.02	806.00	7.07	749.00
Tetratriacontane	889.30	−0.21	887.40	10.45	794.70
Pentatriacontane	914.70	0.13	915.90	10.74	817.50
MAPD		3.07		8.80	

**Table 10 molecules-25-01147-t010:** Comparison of the Cp(sol,298) data of the ionic and the non-ionic forms of amino acids calculated by the present GA and the V_m_ [1] method (in J/mol/K).

Molecule name	C_p_(sol,298) GA-calc.		C_p_(sol,298) Exp.	C_p_(sol,298) V_m_-Calc. [1]	
	Non-Ionic	Zwitter-Ionic		Zwitter-Ionic	Non-Ionic
Glycine	120.20	89.90	99.30	95.90	91.30
Alanine	147.60	116.50	119.90	118.50	116.10
N-Methylglycine	136.00	122.40	118.20	119.10	116.90
Serine	152.70	121.70	135.60	126.20	126.50
Aminobutyric acid	173.00	142.00	146.40	140.90	140.70
Proline	162.20	137.30	150.40	149.80	150.90
Threonine	183.00	152.00	155.31	153.30	154.70
Aspartic acid	193.00	162.00	155.18	155.10	156.00
Asparagine	189.80	158.70	159.80	157.60	159.30
Valine	196.70	165.60	165.00	162.90	165.40
5-Aminopentanoic acid	196.60	166.30	163.70	164.30	166.60
Ornithine	225.40	194.40	191.20	179.20	183.60
Glutamine	214.00	183.00	184.18	182.30	186.50
Leucine	222.10	191.10	200.80	183.90	188.20
Isoleucine	222.10	191.10	188.28	184.70	189.90
Methionine	238.50	207.40	205.16	189.10	194.30
N-Phenylglycine	196.10	162.20	177.40	194.60	198.60
Phenylalanine	232.20	201.10	203.10	215.50	221.60
8-Aminooctanoic acid	273.00	242.70	251.70	232.80	241.90
Tyrosine	248.20	217.10	216.44	236.20	238.10
Tryptophane	268.30	237.20	238.15	252.00	263.90

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
