# Peer review of "Calculation of the Isobaric Heat Capacities of the Liquid and Solid Phase of Organic Compounds at 298.15K by Means of the Group-Additivity Method"

_molecules, 2020, doi:10.3390/molecules25051147_

Round 1
Reviewer 1 Report
This is an incremental work of author’s previous work. Although the topic is of general interest, the presentation of the work is not of high standard. The lack of methodological details indicates questionable reproducibility of data by other researchers. The quality of plots presented in the work is poor. If the editor finds the content of the paper satisfies the journals ethical standard, I recommend publication of the paper upon addressing the following comments:
Title:
1. Too long and verbose. Authors should consider making it short and sweet.
2. 298.15 K cloud be replaced by room temperature.
Abstract:
1. This is not clear to me what author mean by “medium absolute percentage deviation.”
2. I am not sure what author mean by “fast Gauss-Seidel fitting calculus.”
Other:
1. Page 3: “General Procedure” should be replaced by ‘Method” or similar…
2. Page 6: 3.2. “Sources of Heat-Capacity Data”: Data set compilation should be a separate section.
3. No reference provided with “The present study is founded on an object-oriented database of at present 32’086 molecules encompassing pharmaceuticals, plant protection, dyes, ionic liquids, liquid crystals, metal-organics, lab intermediates and many more, stored as geometry-optimized 3-dimensional structures, including – besides several further descriptors – a set of 1202 experimental heat capacities of liquids and a corresponding set of 800 heat capacities of solids.”
4. Author should clearly specify the location of “32086” molecules database with link and reference.
5. Page 5: Authors should provide reference “Normalization of this matrix into an Ax = B matrix and its equalization by means of the Gauss-Seidel calculus yields the atom-group contributions.”
6. Page 5: Author should define what is “equalization procedure.”
Author Response
To the Peer Reviewer,
Please find the reply to your comments as attachment.
R. Naef

Reviewer 2 Report
The comments are attached in the file below.

Author Response

(The authors gave the same response as above.)
